# MolCA: Molecular Graph-Language Modeling with Cross-Modal Projector and Uni-Modal Adapter

**Zhiyuan Liu**[†]   **Sihang Li**[‡]   **Yanchen Luo**[‡]   **Hao Fei**[†]
**Yixin Cao**[§]   **Kenji Kawaguchi**[†]   **Xiang Wang**[‡*]   **Tat-Seng Chua**[†]

[†]National University of Singapore, [‡]University of Science and Technology of China
[§]Singapore Management University

{acharkq,sihang0520,luoyc0830,caoyixin2011,xiangwang1223}@gmail.com
haofei37@nus.edu.sg, {kenji,chuats}@comp.nus.edu.sg

## Abstract

Language Models (LMs) have demonstrated impressive molecule understanding ability on various 1D text-related tasks. However, they inherently lack 2D graph perception — a critical ability of human professionals in comprehending molecules' topological structures. To bridge this gap, we propose **MolCA**: Molecular Graph-Language Modeling with Cross-Modal Projector and Uni-Modal Adapter. MolCA enables an LM (*i.e.,* Galactica) to understand both text- and graph-based molecular contents via the cross-modal projector. Specifically, the cross-modal projector is implemented as a Q-Former to connect a graph encoder's representation space and an LM's text space. Further, MolCA employs a uni-modal adapter (*i.e.,* LoRA) for the LM's efficient adaptation to downstream tasks. Unlike previous studies that couple an LM with a graph encoder via cross-modal contrastive learning, MolCA retains the LM's ability of open-ended text generation and augments it with 2D graph information. To showcase its effectiveness, we extensively benchmark MolCA on tasks of molecule captioning, IUPAC name prediction, and molecule-text retrieval, on which MolCA significantly outperforms the baselines. Our codes and checkpoints can be found at https://github.com/acharkq/MolCA.

## 1 Introduction

Language Models (LMs) have demonstrated significant achievements across various domains (Devlin et al., 2019; Zhao et al., 2023). Notably, the wealth of biochemical literature in LMs' pretraining data has enabled LMs to obtain a high-level understanding of biochemical concepts and molecule properties. This can be reflected by their promising performances in biochemical and medical question-answering benchmarks (Taylor et al., 2022; Ope-

nAI, 2023). Therefore, it becomes increasingly urgent to incorporate these LMs to augment research in chemistry and biology.

For this purpose, we aim to utilize LMs for molecule understanding. As shown in Figure 1a, most existing LMs (Touvron et al., 2023; Zhang et al., 2022; Zeng et al., 2022) represent molecules by their 1D Simplified Molecular Input Line Entry System (SMILES) strings (Weininger, 1988) and process them in a manner similar to texts. While convenient, treating molecules as strings overlooks the molecules' 2D graph representations, which are crucial to human professionals in comprehending the molecule structures (Wells, 2012). To combat that, recent works (Su et al., 2022; Liu et al., 2022b) represent molecules as graphs and use a Graph Neural Network (GNN; Xu et al., 2019) as the molecular graph encoder. The graph encoder is trained jointly with an LM through cross-modal contrastive learning (Radford et al., 2021; Li et al., 2022), as illustrated in Figure 1b. However, the application scope of cross-modal contrastive learning is limited (Alayrac et al., 2022): it is suitable for retrieval tasks, but is insufficient for open-ended molecule-to-text generation tasks, such as molecule captioning (Edwards et al., 2022) and molecule's IUPAC name prediction (Taylor et al., 2022). This is because molecule-to-text generation is a conditional generation task (Keskar et al., 2019; Raffel et al., 2020). It requires the LM to understand 2D graphs as the generation conditions, which contrastive learning cannot achieve. Su et al. (2022) attempt to directly input 2D graphs' representations into LMs, however showing limited improvement.

To bridge this gap, we devise **MolCA**: Molecular Graph-Language Modeling with Cross-Modal Projector and Uni-Modal Adapter. MolCA enables the LM to understand 2D graphs as inputs, therefore effectively conditioning the molecule-to-text gener-

---

*Corresponding author. Xiang Wang is also affiliated with Institute of Artificial Intelligence, Institute of Dataspace, Hefei Comprehensive National Science Center.

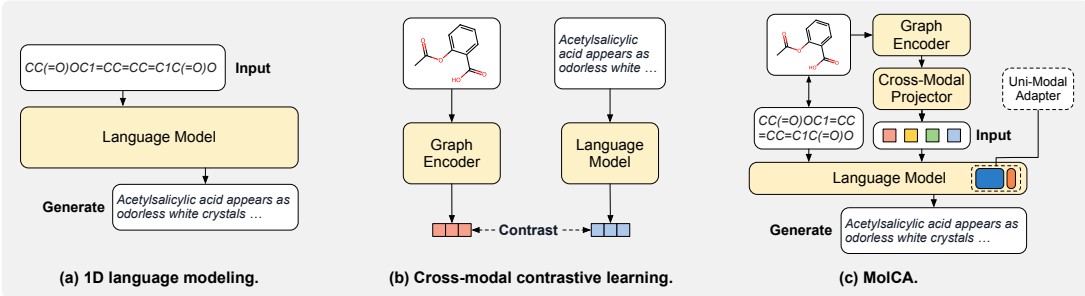

Figure 1: Comparison of molecular language modeling methods.

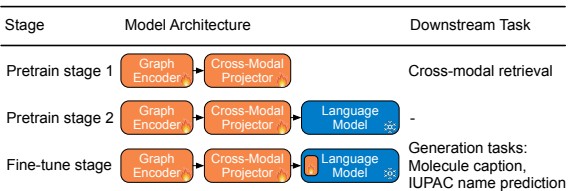

Figure 2: MolCA's three-stage training pipeline.

ation process. To enable the LM to understand 2D graphs, we identify that the key challenge is **cross-modal alignment** (Li et al., 2023; Merullo et al., 2023; Alayrac et al., 2022): translating the representations of 2D graphs into 1D soft prompts (Li and Liang, 2021) in the text space so that the LM can understand. This translation is facilitated by the cross-modal projector, bridging the gap between the graph encoder's representation space and the LM's input space, as illustrated in Figure 1. Specifically, we implement the cross-modal projector as a Q-Former (Li et al., 2023) due to its effectiveness in vision-language tasks. With an effective cross-modal projector, we can harness the power of existing large LMs (Taylor et al., 2022; Touvron et al., 2023) for molecule-to-text generation. However, given a large LM with billion scale parameters, its efficiency of downstream fine-tuning arises as a new problem. Therefore, we integrate the LM with a uni-modal adapter, *i.e.,* LoRA (Hu et al., 2022), to enable its efficient adaptation.

As Figure 2 illustrates, MolCA uses a three-stage training pipeline to integrate its components. The two pretrain stages aim to develop the cross-modal alignment ability of the cross-modal projector. In pretrain stage 1, the projector and the encoder are trained to extract the molecule features that are the most relevant to the text. This stage endows the resulting model with powerful molecule-text retrieval ability. In pretrain stage 2, the cross-modal projector is connected to a frozen LM and trained for molecule captioning. This task forces the cross-modal projector to produce soft prompts that the LM can understand. In the final stage, MolCA is

fine-tuned for downstream generation tasks.

Our contributions can be summarized as follows:

- We propose MolCA, a pioneering method for molecular language modeling. MolCA enables an LM to perceive 2D molecular graphs, thereby facilitating molecule-to-text generation tasks.

- MolCA sets new state-of-the-arts in a variety of benchmarks. It surpasses the baselines by 4.0 and 8.7 BLEU-2 for molecule captioning on CheBI-20 (Edwards et al., 2022) and our curated PubChem324k dataset, respectively. Moreover, in predicting IUPAC names, MolCA shows a significant advantage of 10.0 BLEU-2 over the baselines. For molecule-text retrieval, MolCA outperforms the baselines by 20% retrieval accuracy in PubChem324k and achieves the best performances in PCDes (Zeng et al., 2022) and MoMu datasets (Su et al., 2022).

- We conduct ablation studies to show MolCA's effectiveness of incorporating 2D graphs into LMs for molecule-related tasks. Additionally, our quantitative analysis shows that incorporating 2D graphs helps improve the LM's ability to count functional groups inside molecules.

## 2 Model Architecture

Here we introduce three key components of MolCA's architecture: 1) a graph encoder for 2D structure understanding, 2) an LM for text generation, and 3) a cross-modal projector to connect the graph encoder and the LM. We describe the uni-modal adapter in Section 3.3.

**Graph Encoder.** Given the rich structural patterns in molecules, we leverage a GNN-based encoder to encode molecular graphs. Specifically, we employ a five-layer GINE (Hu et al., 2020) that is pretrained on 2 million molecules from the ZINC15 (Sterling and Irwin, 2015) dataset by contrastive learning (You et al., 2020). Given a molecular graph $g$, the graph encoder $f$ can generate

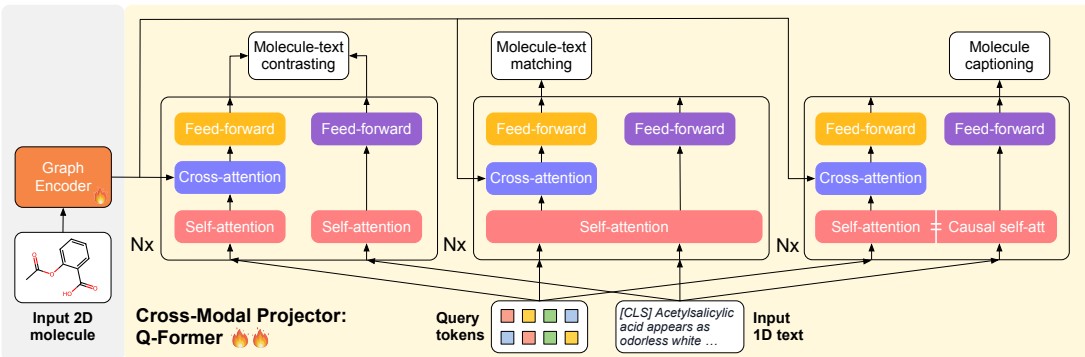

Figure 3: MolCA's pretrain stage 1. The graph encoder and the cross-modal projector (*i.e.,* Q-Former) are jointly optimized using three cross-modal tasks. Modules of the same color share weights.

structure-aware features for every node of $g$:

$$f(g) = \mathbf{Z} \in \mathbb{R}^{|g| \times d}, \qquad (1)$$

where $|g|$ denotes the number of nodes in $g$.

**Language Model.** To achieve effective text generation performance, we employ Galactica (Taylor et al., 2022) as the base LM. Galactica is pretrained on a large collection of scientific literature, which encompasses fields like chemistry, biology, and medicine. Its promising performance in text-based science question-answering benchmarks (Hendrycks et al., 2021; Jin et al., 2019) underscores its understanding of high-level biochemical concepts. Notably, Galactica can process 1D SMILES of molecules, which can potentially benefit our downstream tasks. Galactica is a decoder-only transformer LM based on the OPT (Zhang et al., 2022) architecture.

**Cross-Modal Projector.** We implement the cross-modal projector as a Querying-Transformer (Q-Former) (Li et al., 2023) to map the graph encoder's outputs to the LM's input text space. As shown in Figure 3, Q-former has different procedures for processing 2D molecular graphs and 1D texts. Given text inputs, Q-Former inserts [CLS] tokens at the beginning and processes the texts by N layers of self-attention modules and feed-forward networks. The self-attention modules adopt causal masks (Raffel et al., 2020) when the pretraining task is text generation. On the other hand, given a molecular graph $g$, Q-Former works as a molecule feature extractor. Specifically, it maintains a set of learnable query tokens $\{\boldsymbol{q}_k\}_{k=1}^{N_q}$ as inputs. These query tokens can interact with the graph encoder's output $\mathbf{Z}$ through the cross-attention modules (Vaswani et al., 2017) and extract molecule features. The cross-attention modules are added every two layers. Additionally, the query

tokens can interact with the text inputs through the same self-attention modules. Note that, the query tokens and text inputs are processed by different feed-forward networks, in order to maintain capacities for processing molecules and texts.

We initialize Q-Former from Sci-BERT (Beltagy et al., 2019), an encoder-only transformer pretrained on scientific publications. Q-Former's cross-attention modules are randomly initialized.

## 3 Training Pipeline

This section delves into the details of MolCA's three-stage training pipeline (*cf.* Figure 2). The two pretrain stages leverage a dataset of molecule-text pairs $\mathcal{D} = \{(g_1, \boldsymbol{y}_1), (g_2, \boldsymbol{y}_2), ...\}$ to train the cross-modal projector and the graph encoder. The goal of pretraining is to translate 2D molecular graphs into soft prompts that a frozen LM can understand. The fine-tune stage focuses on efficient adaptation to downstream generation tasks.

### 3.1 Pretrain Stage 1: Learning to Extract Text Relevant Molecule Representations

In this stage, we aim to optimize the cross-modal projector (*i.e.,* Q-Former) to extract the molecule features most relevant to the text input. This stage serves as a "warmup" training for the cross-modal projector before connecting to the LM. Inspired by BLIP2 (Li et al., 2023), we simultaneously apply three cross-modal pretraining tasks that are tailored for Q-Former's architecture: molecule-text contrasting, molecule-text matching, and molecule captioning. These pretraining tasks endow the Q-Former with a strong molecule-text retrieval ability. Therefore, we save the resulting model from this stage for downstream retrieval tasks. We now elaborate on the three pretraining tasks.

**Molecule-Text Contrasting (MTC).** We apply

cross-modal contrastive learning (Radford et al., 2021) to train the Q-Former to extract text-revelant molecule features. In this task, query tokens and text inputs are fed into the Q-Former separately (left of Figure 3) to obtain Q-Former's molecule representations and text representations.

Formally, let $\{(g_1, \boldsymbol{y}_1), ..., (g_B, \boldsymbol{y}_B)\}$ be a batch of molecule-text pairs. We denote $g_i$'s Q-Former representations as $\{\boldsymbol{m}_{ik}\}_{k=1}^{N_q}$ (each element for one query token), and denote $\boldsymbol{y}_i$'s Q-Former representation as $\boldsymbol{t}_i$ (representation of the [CLS] token). For arbitrary $i, j \in [1, B]$, we measure the similarity between $\boldsymbol{t}_i$ and $\{\boldsymbol{m}_{jk}\}_{k=1}^{N_q}$ by computing the maximum similarity between $\boldsymbol{t}_i$ and every element in $\{\boldsymbol{m}_{jk}\}_{k=1}^{N_q}$. The MTC loss $\ell_{\text{MTC}}$ can be written as:

$$\ell_{\text{g2t}} = \sum_{i=1}^{B} \log \frac{\exp(\max_k \cos(\boldsymbol{m}_{ik}, \boldsymbol{t}_i)/\tau)}{\sum_{j=1}^{B} \exp(\max_k \cos(\boldsymbol{m}_{ik}, \boldsymbol{t}_j)/\tau)},$$

$$\ell_{\text{t2g}} = \sum_{i=1}^{B} \log \frac{\exp(\max_k \cos(\boldsymbol{t}_i, \boldsymbol{m}_{ik})/\tau)}{\sum_{j=1}^{B} \exp(\max_k \cos(\boldsymbol{t}_i, \boldsymbol{m}_{jk})/\tau)},$$

$$\ell_{\text{MTC}} = -\frac{1}{B}\ell_{\text{g2t}} - \frac{1}{B}\ell_{\text{t2g}}, \qquad (2)$$

where $\cos(\cdot, \cdot)/\tau$ is the temperature-scaled cosine similarity. Temperature $\tau$ is empirically set to $0.1$.

**Molecule-Text Matching (MTM).** MTM is a binary classification task, aiming to predict whether a molecule-text pair is matched (positive) or unmatched (negative). As Figure 3 illustrates, MTM allows the queries and the texts to interact through the same self-attention module. In this way, the queries can extract multi-modal information from both molecules and texts. For MTM prediction, we attach a linear classifier after the mean pooling of all queries' Q-Former representations. Let $\rho(g, \boldsymbol{y})$ denotes MTM's predicted probability that $(g, \boldsymbol{y})$ is matched. MTM loss $\ell_{\text{MTM}}$ can be written as:

$$\ell_{\text{MTM}} = \frac{1}{B} \mathbb{E}_{j,k \sim \text{U}(1,B)} \Big[ \sum_{i=1}^{B} -\log \rho(g_i, \boldsymbol{y}_i) +$$

$$\log \rho(g_i, \boldsymbol{y}_j) + \log \rho(g_k, \boldsymbol{y}_i) \Big], \qquad (3)$$

where $\text{U}(1, B)$ is a uniform distribution; $\boldsymbol{y}_j$ and $g_k$ are random negative samples in batch.

Similar to MTC, MTM also computes the similarity between molecule-text pairs. The difference is that MTM can capture more fine-grained similarity between a molecule and a text through the self-attention and cross-attention modules, compared to the simple cosine similarity used by MTC.

Therefore, in retrieval experiments, we use MTC to first retrieve the top k samples and use MTM for re-ranking, thereby improving the performance.

**Molecule Captioning (MCap).** MCap aims to generate the molecule's text description based on the molecule representations. For this task, we adopt a special masking strategy in self-attention modules to ensure that the queries learn to extract molecule features that correspond to the text descriptions. Specifically, we employ the bidirectional self-attention masks for queries, allowing them to see each other but not the text tokens. Further, we apply causal masks for texts on the same self-attention module to perform autoregressive decoding of text descriptions. Each text token can see the queries and the preceding text, but not the subsequent text tokens. Since the text tokens cannot directly interact with the graph encoder, they must obtain molecule information from the queries, forcing the queries to extract molecule information through the cross-attention modules. Let $p_1(\boldsymbol{y}|g)$ be the probability of Q-Former generating text $\boldsymbol{y}$ for a graph $g$. We use the following loss function:

$$\ell_{\text{MCap}} = -\frac{1}{B} \sum_{i=1}^{B} \log p_1(\boldsymbol{y}_i | g_i) \qquad (4)$$

## 3.2 Pretrain Stage 2: Aligning 2D Molecular Graphs to Texts via Language Modeling

In this stage, we aim to align the cross-modal projector's outputs to the text space of a frozen LM. As Figure 4 illustrates, we feed the cross-modal projector's representations of 2D molecular graphs to the frozen LM as inputs, and train the model to generate molecules' text descriptions. This process encourages the cross-modal projector to provide representations that the LM can understand, so as to prompt the text generation. Additionally, we also use a molecule's 1D SMILES to guide the generation (*cf.* Figure 4). This is because most LMs (Taylor et al., 2022; Touvron et al., 2023; Zhang et al., 2022) use SMILES during pretraining. Therefore, these LMs have established some correlations between SMILES and their text contexts. Thus, including SMILES can potentially prompt the corresponding biochemical knowledge. On the other hand, incorporating 2D graphs can help capture structural patterns that are hard to learn from 1D SMILES. We will show later in experiments that combining 2D graphs and 1D SMILES can boost performance.

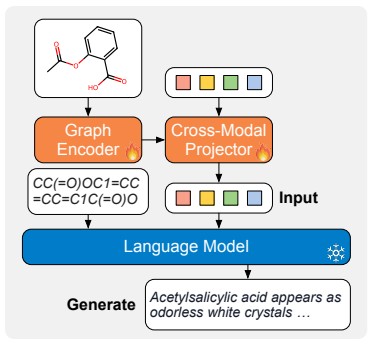

Figure 4: MolCA's pretrain stage 2 by molecule captioning.

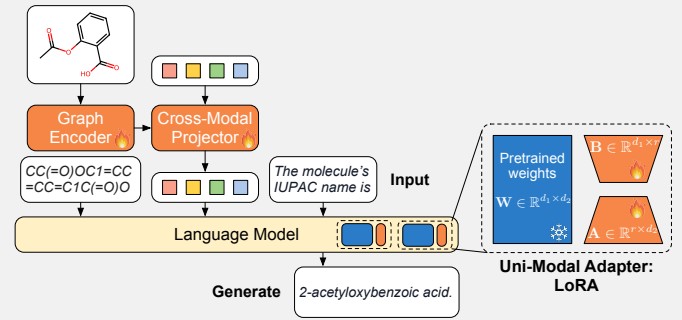

Figure 5: MolCA's fine-tune stage for molecule-to-text generation. The example shows the prediction of a molecule's IUPAC name.

Formally, consider a molecule-text pair $(g, \boldsymbol{y})$ and $g$'s SMILES repsentation $\boldsymbol{s}$, The cross-modal projector representations of $g$ are denoted as $\{\boldsymbol{m}_k\}_{k=1}^{N_q}$. We define $p_2(\cdot)$ as the text distribution parameterized by the frozen LM. We optimize the cross-modal projector and the graph encoder by minimizing the following loss function:

$$-\log p_2(\boldsymbol{y}|\{\boldsymbol{m}_k\}_{k=1}^{N_q}, \boldsymbol{s})$$
$$= -\sum_{l=1}^{L} \log p_2(y_l|y_1, ..., y_{l-1}, \{\boldsymbol{m}_k\}_{k=1}^{N_q}, \boldsymbol{s}). \quad (5)$$

### 3.3 Fine-tune Stage: Uni-Modal Adapter for Efficient Downstream Adaptation

In this stage, we fine-tune MolCA for downstream generation tasks. As Figure 5 illustrates, we append a text prompt of the task description after the molecule representations. Then, we apply language modeling loss to fine-tune MolCA for generation tasks, such as molecule's IUPAC name prediction.

**Uni-Modal Adapter.** In MolCA, the LM is accounted for a large portion of computation overhead: it can have $\sim$1B parameters, while the cross-modal projector and graph encoder only have a total of $\sim$0.1B parameters. Therefore, we employ a uni-modal adapter for the LM's efficient adaptation to downstream tasks. Specifically, we employ the LoRA (Hu et al., 2022) adapter due to its simple implementation and promising performances (Liu et al., 2022a). As shown in Figure 5, for selected weight matrices (*e.g.*, $\mathbf{W} \in \mathbb{R}^{d_1 \times d_2}$) in the LM, LoRA adds pairs of rank decomposition matrices (*e.g.*, $\mathbf{BA}, \mathbf{B} \in \mathbb{R}^{d_1 \times r}, \mathbf{A} \in \mathbb{R}^{r \times d_2}$) in parallel to them. The original $\boldsymbol{h} = \mathbf{W}\boldsymbol{x}$ layer is changed to:

$$\boldsymbol{h} = \mathbf{W}\boldsymbol{x} + \mathbf{BA}\boldsymbol{x}, \quad (6)$$

where $\mathbf{W}$ is kept frozen and the newly added $\mathbf{BA}$ is trained during adaptation. Given a small $r \ll$

| Subset | Size | Avg mol len | Min text len | Avg text len |
|--------|------|-------------|--------------|--------------|
| Pretrain | 309689 | 35 | 1 | 18 |
| Train | 12000 | 32 | 20 | 60 |
| Valid | 1000 | 32 | 20 | 61 |
| Test | 2000 | 31 | 20 | 60 |

Table 1: Statistics of the PubChem324k dataset. We count the text length by splitting the text at spaces.

$\min(d_1, d_2)$, LoRA can effectively adapt the LM to downstream tasks while requiring little memory overhead for storing gradients.

## 4 Experiments

### 4.1 Experimental Setting

Here we briefly present the experimental settings. More details can be found in Appendix B.

**PubChem324k Dataset.** We collect PubChem-324k – a dataset containing 324k molecule-text pairs from the PubChem website[1]. Table 1 presents the dataset statistics. Notice that, the dataset includes many uninformative texts, such as "The molecule is a peptide". Therefore, we sample a high-quality subset of 15k pairs with text longer than 19 words for downstream tasks. This high-quality subset is further randomly divided into the train/valid/test sets. The remaining dataset, which is more noisy, is used for pretraining.

**Baselines.** For generation tasks, we compare MolCA with the following baselines: T5 (Raffel et al., 2020), MolT5 (Edwards et al., 2022), and MoMu (Su et al., 2022). For molecule-text retrieval, we also include these methods: MoleculeSTM (Liu et al., 2022b), KV-PLM (Zeng et al., 2022), and Sci-BERT (Beltagy et al., 2019).

### 4.2 Molecule Captioning

We evaluate MolCA for molecule captioning on the datasets of PubChem324k and CheBI-20 (Edwards

---

[1] https://pubchem.ncbi.nlm.nih.gov

| Model | #Trainable params | BLEU-2 | BLEU-4 | ROUGE-1 | ROUGE-2 | ROUGE-L | METEOR |
|---|---|---|---|---|---|---|---|
| **1D SMILES** | | | | | | | |
| MolT5-Small | 80M, full ft | 14.8 | 8.5 | 26.5 | 13.5 | 23.6 | 18.5 |
| MolT5-Base | 250M, full ft | 30.1 | 20.9 | 40.3 | 25.1 | 33.8 | 35.6 |
| MolT5-Large | 780M, full ft | 30.2 | 22.2 | 41.5 | 25.9 | 34.8 | 36.6 |
| **1D SMILES + 2D Graph** | | | | | | | |
| MoMu-Small | 82M, full ft | 19.1 | 12.0 | 29.7 | 16.3 | 26.7 | 21.8 |
| MoMu-Base | 252M, full ft | 30.2 | 21.5 | 40.5 | 25.1 | 34.4 | 34.2 |
| MoMu-Large | 782M, full ft | 31.1 | 22.8 | 41.8 | 25.7 | 36.7 | 36.2 |
| MolCA, MolT5-Large | 877M, full ft | 33.7 | 27.0 | 49.7 | 35.6 | 44.4 | 42.4 |
| MolCA, Galac$_{125M}$ | 222M, full ft | 32.4 | 24.9 | 44.9 | 30.1 | 39.5 | 39.2 |
| MolCA, Galac$_{1.3B}$ | 100M, LoRA ft* | **39.8** | **31.7** | **51.7** | **37.3** | **46.2** | **46.8** |

(a) PubChem324k dataset. Baseline performances are reproduced using their source codes (Edwards et al., 2022; Su et al., 2022).

| Model | #Trainable params | BLEU-2 | BLEU-4 | ROUGE-1 | ROUGE-2 | ROUGE-L | METEOR |
|---|---|---|---|---|---|---|---|
| **1D SMILES** | | | | | | | |
| T5-Small | 80M, full ft | 50.1 | 41.5 | 60.2 | 44.6 | 54.5 | 53.2 |
| T5-Base | 250M, full ft | 51.1 | 42.3 | 60.7 | 45.1 | 55.0 | 53.9 |
| T5-Large | 780M, full ft | 55.8 | 46.7 | 63.0 | 47.8 | 56.9 | 58.6 |
| MolT5-Small | 80M, full ft | 51.9 | 43.6 | 62.0 | 46.9 | 56.3 | 55.1 |
| MolT5-Base | 250M, full ft | 54.0 | 45.7 | 63.4 | 48.5 | 57.8 | 56.9 |
| MolT5-Large | 780M, full ft | 59.4 | 50.8 | 65.4 | 51.0 | 59.4 | 61.4 |
| **1D SMILES + 2D Graph** | | | | | | | |
| MoMu-Small | 82M, full ft | 53.2 | 44.5 | - | - | 56.4 | 55.7 |
| MoMu-Base | 252M, full ft | 54.9 | 46.2 | - | - | 57.5 | 57.6 |
| MoMu-Large | 782M, full ft | 59.9 | 51.5 | - | - | 59.3 | 59.7 |
| MolCA, Galac$_{125M}$ | 222M, full ft | 61.6 | 52.9 | 67.4 | 53.3 | 61.5 | 63.9 |
| MolCA, Galac$_{1.3B}$ | 110M, LoRA ft* | **63.9** | **55.5** | **69.7** | **55.8** | **63.6** | **66.9** |

(b) CheBI-20 dataset. Baseline performances are borrowed from their original papers (Edwards et al., 2022; Su et al., 2022).

Table 2: Performances (%) of molecule captioning on the PubChem324k and CheBI-20 datasets. **Bold** indicates the best performance and underline indicates the second best performance. Full ft denotes full parameter fine-tuning. *The LoRA configurations for PubChem324k and CheBI-20 datasets are different. Details are in Appendix B.

et al., 2022). Specifically, we implement MolCA with the base LMs of Galactica$_{1.3B}$, Galactica$_{125M}$, and MolT5-Large. We employ full parameter fine-tuning for Galactica$_{125M}$ and MolT5-Large due to their smaller scales. We fine-tune MolCA and baselines on the dataset's training set and report the test set performance selected by the valid set. Following (Edwards et al., 2022), we adopt BLEU (Papineni et al., 2002), ROUGE (Lin, 2004), and METEOR (Banerjee and Lavie, 2005) as the evaluation metrics. As shown in Table 2, we observe that:

1. MolCA consistently outperforms the baselines by a large margin. Specifcally, MolCA, Galac$_{1.3B}$ achieves the highest performance on all metrics. It outperforms the baselines by 8.7 BLEU-2 on PubChem324k and 4.0 BLEU-2 on CheBI-20.

2. MolCA, Galac$_{125M}$ outperforms baselines of larger sizes across all metrics, showing that MolCA's advantage is not limited to model scale.

### 4.3 IUPAC Name Prediction

The International Union of Pure and Applied Chemistry (IUPAC) has established a standardized nam-

ing system for chemical compounds, known as IUPAC names (Favre and Powell, 2013). Notably, this naming system relies on identifying specific molecule structures, including hydrocarbon chains and double/triple bonds. Therefore, correctly predicting IUPAC names indicates a model's proficiency to understand molecule structures. We fine-tune MolCA and baselines using the PubChem324k's training set to generate a molecule's IUPAC name. As shown in Table 3, MolCA consistently outperforms the baselines by a large margin of 10.0 BLEU-2, highlighting MolCA's advantage in comprehending molecule structures.

### 4.4 Molecule-Text Retrieval

We evaluate MolCA for molecule-text retrieval on the datasets of PubChem324k, PCDes (Zeng et al., 2022) and MoMu (Su et al., 2022). Specifically, we evaluate MolCA's checkpoint from pretrain stage 1 without further fine-tuning. For all experiments, MolCA first retrieves the top 128 candidates using MTC, then employs the MTM module for reranking. We select Accuracy (Acc) and Recall@20

| Model | #Trainable params | BLEU-2 | BLEU-4 | ROUGE-1 | ROUGE-2 | ROUGE-L | METEOR |
|---|---|---|---|---|---|---|---|
| **1D SMILES** | | | | | | | |
| MolT5-Small | 80M, full ft | 48.6 | 35.2 | 40.0 | 16.1 | 34.3 | 42.5 |
| MolT5-Base | 250M, full ft | 52.7 | 41.5 | 50.7 | 26.0 | 44.3 | 53.2 |
| MolT5-Large | 780M, full ft | 59.4 | 49.7 | 55.9 | 33.3 | 49.1 | 58.5 |
| **1D SMILES + 2D Graph** | | | | | | | |
| MolCA, Galac$_{125M}$ | 222M, full ft | 72.9 | 65.1 | 69.5 | 48.0 | 62.6 | 71.6 |
| MolCA, Galac$_{1.3B}$ | 100M, LoRA ft | **74.6** | **66.1** | **70.5** | **49.1** | **64.2** | **73.0** |

Table 3: Performances (%) of predicting molecule's IUPAC names on the PubChem324k dataset. Baseline performances are obtained by running their source codes (Edwards et al., 2022).

| | M2T | | T2M | |
|---|---|---|---|---|
| Model | Acc | R@20 | Acc | R@20 |
| **1D SMILES** | | | | |
| Sci-BERT | 39.3 | 86.1 | 37.9 | 85.1 |
| KV-PLM | 38.8 | 86.3 | 38.7 | 85.6 |
| **2D Graph** | | | | |
| MoMu-S* | 11.5 | 41.2 | 12.6 | 43.6 |
| MoMu-K* | 11.3 | 41.0 | 12.4 | 39.9 |
| MoMu-S | 40.6 | 86.5 | 40.6 | 86.5 |
| MoMu-K | 41.8 | 88.9 | 42.4 | 88.5 |
| MoleculeSTM | 47.1 | 89.0 | 45.4 | 91.5 |
| MolCA w/o MTM | 60.5 | 93.7 | 58.6 | 92.3 |
| MolCA | **69.4** | **95.7** | **69.6** | **94.6** |

(a) Performances (%) in the PubChem324k dataset.

| | PCDes dataset | | MoMu dataset | |
|---|---|---|---|---|
| Model | M2T | T2M | M2T | T2M |
| **1D SMILES** | | | | |
| Sci-BERT[†] | 60.7 | 60.8 | 0.3 | 0.3 |
| KV-PLM[†] | 75.9 | 64.3 | 0.5 | 0.3 |
| **2D Graph** | | | | |
| MoMu-S[†] | 79.1 | 75.5 | 43.3 | 43.4 |
| MoMu-K[†] | 80.2 | 79.0 | 43.7 | 43.5 |
| MoleculeSTM | 84.6 | 85.1 | 75.8 | 74.5 |
| MolCA w/o MTM | 88.0 | 85.5 | 81.5 | 81.6 |
| MolCA | **90.5** | **87.6** | **88.6** | **87.3** |

(b) Recall@20 (%) in the PCDes and MoMu datasets.

Table 4: Molecule-text retrieval performances. We report performances of using molecule to retrieve text (M2T) and using text to retrieve molecule (T2M). * denotes performance evaluated on the baseline's released checkpoint. † denotes result borrowed from (Su et al., 2022). Other models are trained on PubChem324k's pretrain subset. The complete results are in Appendix C

(R@20) as the evaluation metrics, and report the performances of retrieval in the entire test set. As shown in Table 4, we observe that:

1. MolCA demonstrates superior performance over baselines. Specifically, in PubChem324k, MolCA improves the accuracy by more than 20% over the baselines. In PCDes and MoMu, MolCA also consistently outperforms the baselines, demonstrating its effectiveness for molecule-text retrieval.

2. Incorporating MTM significantly improves

MolCA's performance. This can be attributed to MTM's ability to model long-range interactions between molecule features and texts, achieved by the cross-attention and self-attention modules.

3. MolCA's good performances can be partially attributed to our larger pretrain dataset – PubChem324k. As shown in Table 4a, we compare the performances of MoMu's original checkpoint (pretrained on 15k molecule-text pairs) with our reproduced MoMu using PubChem324k. The latter improves the retrieval accuracy by over 25%.

### 4.5 Ablation Study on Representation Types

Here we ablate the two representations types of molecules: 1D SMILES and 2D graphs. We compare MolCA with its two variants: 1) 1D SMILES: an LM that uses only 1D SMILES for pretraining and fine-tuning. For a fair comparison, we pretrain this variant on PubChem324k's pretrain subset for molecule captioning before its downstream adaptation; 2) 2D Graph: this variant follows the original MolCA's training pipeline, except not using 1D SMILES in pretrain stage 2 and fine-tune stage.

**End Task Ablation.** Table 5 presents the results for molecule-to-text generation and molecule property prediction (Hu et al., 2020) tasks. We can observe that combing 2D graphs and 1D SMILES leads to improved performance in all the compared tasks. This demonstrates MolCA's effectiveness in incorporating molecules' 2D graph representations.

**Counting Functional Groups (FGs).** We ablate MolCA's capability of counting 85 types of FGs inside molecules. An FG is a molecule's subgraph that exhibits consistent chemical behaviors across different molecules (Rong et al., 2020). Correctly counting FGs can help understand a molecule's properties. As shown in Figure 6, incorporating 2D graphs significantly improves MolCA's performance in counting FGs, thereby enhancing its ability in understanding molecule structures.

| Representation type | BLEU-2 | BLEU-4 | ROUGE-1 | ROUGE-2 | ROUGE-L | METEOR |
|---|---|---|---|---|---|---|
| **Molecule Captioning, PubChem324k** | | | | | | |
| 1D SMILES | 33.7 | 26.0 | 45.4 | 31.6 | 40.7 | 40.3 |
| 2D Graph | 35.7 | 27.4 | 47.3 | 32.3 | 41.8 | 42.0 |
| 1D SMILES + 2D Graph | **39.8** | **31.7** | **51.7** | **37.3** | **46.2** | **46.8** |
| **Molecule Captioning, CheBI-20** | | | | | | |
| 1D SMILES | 58.3 | 49.4 | 65.9 | 51.3 | 59.7 | 62.4 |
| 1D SMILES + 2D Graph | **63.9** | **55.5** | **69.7** | **55.8** | **63.6** | **66.9** |
| **IUPAC Name Prediction, PubChem324k** | | | | | | |
| 1D SMILES | 70.7 | 60.7 | 68.6 | 46.2 | 61.7 | 71.5 |
| 1D SMILES + 2D Graph | **74.6** | **66.1** | **70.5** | **49.1** | **64.2** | **73.0** |

(a) Ablating the representation type on tasks of molecule captioning and IUPAC name prediction.

| Representation type | Bace | BBBP | ClinTox | ToxCast | Sider | Tox21 | Mean |
|---|---|---|---|---|---|---|---|
| 1D SMILES | 79.3±0.8 | **70.8±0.6** | 89.0±1.7 | 56.2±0.7 | 61.1±1.2 | 76.0±0.5 | 72.1 |
| 1D SMILES + 2D Graph | **79.8±0.5** | 70.0±0.5 | **89.5±0.7** | **64.5±0.8** | **63.0±1.7** | **77.2±0.5** | **74.0** |

(b) ROC-AUC (%) scores on six molecule property prediction datasets from MoleculeNet (Wu et al., 2018). We use scaffold split following (Hu et al., 2020). We report the performance's mean values and standard deviations across three random seeds.

Table 5: Ablating molecule's representation types. All compared models fine-tune the base LM of Galactica$_{1.3B}$.

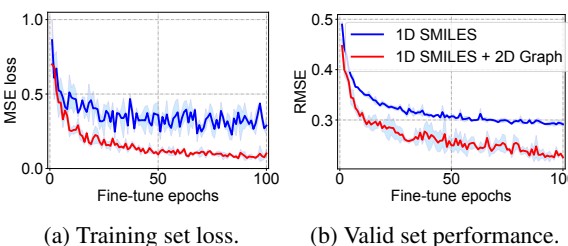

(a) Training set loss.  (b) Valid set performance.

Figure 6: Ablating MolCA for counting FGs inside molecules from PubChem324k. We plot average values across three random seeds. Blue shades indicate the range of ± one standard deviation. Evaluation metric is Root Mean Square Error (RMSE). Lower value indicates better performance.

# 5 Related Works

Here we briefly review the molecule-related literature. We discuss MolCA's relations to vision-language pretraining methods in Appendix A.

**Molecule Understanding via 1D Language Modeling.** Due to the extensive biochemical literature in their training corpus, some open-domain LMs (Zhang et al., 2022; Touvron et al., 2023; Chowdhery et al., 2022) have obtained a high-level understanding of molecular and chemical concepts. This is demonstrated through their promising performances in text-related biochemical and medical question-answering benchmarks (Hendrycks et al., 2021; Jin et al., 2019). Among these LMs, Galactica (Taylor et al., 2022) shows competitive performances for using a corpus that is primarily composed of scientific literature. Focusing on the chemistry domain, KV-PLM (Zeng et al., 2022) models molecules by applying masked language modeling

loss on 1D SMILES. Vaucher et al. (2021) propose to predict the chemistry experiment actions by reading chemical reaction equations. MolT5 (Edwards et al., 2022) presents several T5-based (Raffel et al., 2020) LMs for SMILES-to-text and text-to-SMILES translations. Further, Christofidellis et al. (2023) propose to fine-tune T5 for chemical reaction prediction and retrosynthesis tasks. MolCA is different from these methods that exclusively utilize 1D SMILES to represent molecules. Instead, MolCA aims to enable LMs to perceive molecules' 2D graph representations.

**Molecule-Text Contrastive Learning.** Driven by the demand of a molecule-text retrieval system, Text2Mol (Edwards et al., 2021) employs cross-modal contrastive learning to train a molecular graph encoder of GCNs (Kipf and Welling, 2017) and a text encoder of Sci-BERT (Beltagy et al., 2019). Subsequent works (Su et al., 2022; Liu et al., 2022b; Seidl et al., 2023) have proposed enhancements, including the addition of inter-modal contrastive learning loss (Su et al., 2022) and applying the model for text-based molecule editing (Liu et al., 2022b). However, cross-modal contrastive learning is unsuitable for open-ended conditional generation task (Alayrac et al., 2022), because of its focus on learning a similarity function. To resolve the problem, we propose MolCA to enable the LM's understanding of 2D molecular graphs, facilitating MolCA's capability of open-ended molecule-to-text generation.

# 6 Conclusion and Future Works

In this work, we propose MolCA, a novel molecular language modeling method. MolCA aims to enable LMs to perceive 2D graphs for molecule-to-text generation. For this purpose, MolCA features a cross-modal projector to map representations of 2D graphs into the text space of LMs. It also employs a uni-modal adapter for efficient downstream adaptation. MolCA achieves state-of-the-art performances on molecule captioning and molecule-text retrieval benchmarks. Looking forward, we are interested in exploring LMs for 3D molecular modeling and drug discovery tasks.

## Limitations

This work focuses on utilizing LMs' generation ability for molecule-text tasks. Other interesting abilities of LMs, like in-context learning and chain-of-thought reasoning, are beyond the scope of this research. We leave that to future exploration.

While MolCA offers improvements over baselines, we observe that the current performance in molecule captioning is not yet sufficient for practical application. This can be attributed to the scale of pretraining data. To our knowledge, our PubChem324k dataset is the largest dataset of molecule-text pairs. However, compared to the ∼10M scale dataset (Changpinyo et al., 2021) for vision-language pretraining, our dataset, consists of 324k data points, is comparatively smaller and limits the model's performance. Remedy solutions may include mining weakly supervised data from biochemical literature.

## Broader Impacts

Our work has established new state-of-the-art performances in molecule captioning and molecule-text retrieval. It has broader impacts in two aspects: 1) for chemistry professionals, our method of molecule captioning and molecule-text retrieval could be useful tools, potentially speeding up their research process; 2) for individuals without specialized chemistry knowledge, our method could provide a more affordable way to access the basic chemical information of molecules.

Our model shares the risks of most LMs. It can generate inaccurate information and can potentially be abused to produce biased content. Further, considering the limited scale of our training data, we strongly advise strictly testing our model before applying it in real applications.

## Acknowledgement

This research is supported by NExT Research Center. This material is based upon work supported by the Google Cloud Research Credit program with the award (6NW8-CF7K-3AG4-1WH1). This research is supported by the National Natural Science Foundation of China (9227010114) and the University Synergy Innovation Program of Anhui Province (GXXT-2022-040).

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

## A Complete Related Works

We present the complete literature review. In addition to the molecule-related literature, as addressed in the main body of the paper, we also discuss MolCA's relation to vision-language pretraining.

**Molecule Understanding via 1D Language Modeling.** Due to the extensive biochemical literature in their training corpus, some open-domain LMs (Zhang et al., 2022; Touvron et al., 2023; Chowdhery et al., 2022) have obtained a high-level understanding of molecular and chemical concepts. This is demonstrated through their promising performances in text-related biochemical and medical question-answering benchmarks (Hendrycks et al., 2021; Jin et al., 2019). Among these LMs, Galactica (Taylor et al., 2022) shows competitive performances for using a corpus that is primarily composed of scientific literature. Focusing on the chemistry domain, KV-PLM (Zeng et al., 2022) models molecules by applying masked language modeling loss on 1D SMILES. Vaucher et al. (2021) propose to predict the chemistry experiment actions by reading chemical reaction equations. MolT5 (Edwards et al., 2022) presents several T5-based (Raffel et al., 2020) LMs for SMILES-to-text and text-to-SMILES translations. Further, Christofidellis et al. (2023) propose to fine-tune T5 for chemical reaction prediction and retrosynthesis tasks. MolCA is different from these methods that exclusively utilize 1D SMILES to represent molecules. Instead, MolCA aims to enable LMs to perceive molecules' 2D graph representations.

**Molecule-Text Contrastive Learning.** Driven by the demand of a molecule-text retrieval system, Text2Mol (Edwards et al., 2021) employs cross-modal contrastive learning to train a molecular graph encoder of GCNs (Kipf and Welling, 2017) and a text encoder of Sci-BERT (Beltagy et al., 2019). Subsequent works (Su et al., 2022; Liu et al., 2022b; Seidl et al., 2023) have proposed improvements, including the addition of inter-modal contrastive learning loss (Su et al., 2022) and applying the model for text-based molecule editing (Liu et al., 2022b). However, cross-modal contrastive learning is unsuitable for open-ended conditional generation task (Alayrac et al., 2022), because of its focus on learning a similarity function. To resolve the problem, we propose MolCA to enable the LM's understanding of 2D molecular graphs, facilitating MolCA's capability of open-ended molecule-to-text generation.

**Vision-Language Pretraining (VLP).** Both VLP and Molecular Language Modeling aim to bridge the gap between text and another modality. Notably, VLP methods of CLIP (Radford et al., 2021) and others (Li et al., 2022; Yao et al., 2022) use contrastive learning to connect a visual encoder and a text encoder. These methods can be applied for tasks like image-text retrieval and zero-shot image classification. Recently, a series of VLP works (Tsimpoukelli et al., 2021; Merullo et al., 2023; Li et al., 2023; Alayrac et al., 2022) show that visual features can be aligned to the text space of LMs. This cross-modal alignment allows LMs to utilize their language generation and few-shot learning abilities for multi-modal tasks. MolCA draws inspiration from these findings. To the best of our knowledge, we are the first to align 2D molecular graphs to the text space of LMs. Furthermore, we incorporate a uni-modal adapter to improve the adaptation efficiency on downstream tasks.

## B Experimental Settings

**Pretrain Settings.** MolCA's pretrain stage 1 has 50 epochs and pretrain stage 2 has 10 epochs. Q-Former has 8 query tokens ($N_q = 8$). Our optimizer's configuration follows (Li et al., 2023). We use the AdamW optimizer (Loshchilov and Hutter, 2019) with a weight-decay of 0.05. The learning rate is scheduled by a combination of linear warmup and cosine decay. The peak learning rate is 1e-4 and the warmup has 1000 steps.

**Molecule Captioning.** MolCA is fine-tuned for 100 epochs using the same configuration of optimizer and learning rate scheduler. LoRA is implemented using the OpenDelta library (Ding et al., 2022) and the PEFT library (Mangrulkar et al., 2022). For the PubChem324k dataset, we set LoRA's rank $r$ to 8 and apply LoRA to Galactica's modules of [q_proj, v_proj]. This configuration yields a LoRA adapter with 2M parameters, which constitutes 0.12% of the parameters in the Galactica$_{1.3B}$. For the CheBI-20 dataset, we set LoRA's rank $r$ to 16 and apply LoRA to Galactica's modules of [q_proj, v_proj, out_proj, fc1, fc2]. This configuration yields a LoRA adapter with 12M parameters, which constitutes 0.94% of the parameters in the Galactica$_{1.3B}$.

**IUPAC Name Prediction.** We collect IUPAC names for molecules in the train/valid/test sets of PubChem324k using the PubChemPy library[2]. The

---

[2] https://github.com/mcs07/PubChemPy

| Subset | #Mol-text pairs | Usage | Avg mol len | Avg text len | Min text len | Max text len |
|--------|-----------------|-------|-------------|--------------|--------------|--------------|
| Pretrain | 309689 | Pretrain stage 1 & 2 | 35 | 18 | 1 | 1305 |
| Train | 12000 | Downstream fine-tune | 32 | 60 | 20 | 937 |
| Valid | 1000 | Downstream validation | 32 | 61 | 20 | 1197 |
| Test | 2000 | Downstream test | 31 | 60 | 20 | 879 |

Table 6: Statistics of the PubChem324k dataset.

| | Retrieval in batch | | | | Retrieval in test set | | | |
| | M2T (%) | | T2M (%) | | M2T (%) | | T2M (%) | |
| Model | Acc | R@20 | Acc | R@20 | Acc | R@20 | Acc | R@20 |
|-------|-----|------|-----|------|-----|------|-----|------|
| **1D SMILES** | | | | | | | | |
| Sci-BERT | 84.1 | 98.6 | 82.5 | 98.2 | 39.3 | 86.1 | 37.9 | 85.1 |
| KV-PLM | 84.3 | 98.3 | 82.3 | 98.3 | 38.8 | 86.3 | 38.7 | 85.6 |
| **2D Graph** | | | | | | | | |
| MoMu-S* | 42.3 | 90.1 | 43.7 | 90.1 | 11.5 | 41.2 | 12.6 | 43.6 |
| MoMu-K* | 43.3 | 90.4 | 45.8 | 89.0 | 11.3 | 41.0 | 12.4 | 39.9 |
| MoMu-S | 82.9 | 99.0 | 83.0 | 99.0 | 40.6 | 86.5 | 40.6 | 86.5 |
| MoMu-K | 84.1 | 98.7 | 83.6 | 98.9 | 41.8 | 88.9 | 42.4 | 88.5 |
| MoleculeSTM | 87.2 | 98.5 | 86.7 | 98.3 | 47.1 | 89.0 | 45.4 | 91.5 |
| MolCA w/o MTM | 86.6 | 98.9 | 85.3 | 98.7 | 60.5 | 93.7 | 58.6 | 92.3 |
| MolCA | **91.4** | **99.9** | **90.1** | **99.2** | **69.4** | **95.7** | **69.6** | **94.6** |

(a) Molcule-text retrieval performances in the PubChem324k dataset.

| | Retrieval in batch | | | | Retrieval in test set | | | |
| | M2T (%) | | T2M (%) | | M2T (%) | | T2M (%) | |
| Model | Acc | R@20 | Acc | R@20 | Acc | R@20 | Acc | R@20 |
|-------|-----|------|-----|------|-----|------|-----|------|
| **1D SMILES** | | | | | | | | |
| Sci-BERT[†] | 62.6 | - | 61.8 | - | - | 60.7 | - | 60.8 |
| KV-PLM[†] | 77.9 | - | 65.0 | - | - | 75.9 | - | 64.3 |
| **2D Graph** | | | | | | | | |
| MoMu-S[†] | 80.6 | - | 77.0 | - | - | 79.1 | - | 75.5 |
| MoMu-K[†] | 81.1 | - | 80.2 | - | - | 80.2 | - | 79.0 |
| MoleculeSTM | 86.2 | - | 83.9 | - | - | 84.6 | - | 85.1 |
| MolCA w/o MTM | 88.4 | 98.8 | 85.5 | 98.5 | 54.6 | 88.0 | 51.8 | 85.5 |
| MolCA | **91.4** | **99.8** | **88.4** | **98.8** | **60.3** | **90.5** | **59.7** | **87.6** |

(b) Molecule-text retrieval performances in the PCDes dataset.

| | Retrieval in batch | | | | Retrieval in test set | | | |
| | M2T (%) | | T2M (%) | | M2T (%) | | T2M (%) | |
| Model | Acc | R@20 | Acc | R@20 | Acc | R@20 | Acc | R@20 |
|-------|-----|------|-----|------|-----|------|-----|------|
| **1D SMILES** | | | | | | | | |
| Sci-BERT[†] | 1.4 | - | 1.6 | - | - | 0.3 | - | 0.3 |
| KV-PLM[†] | 1.5 | - | 1.3 | - | - | 0.5 | - | 0.3 |
| **2D Graph** | | | | | | | | |
| MoMu-S[†] | 45.7 | - | 40.0 | - | - | 43.3 | - | 43.4 |
| MoMu-K[†] | 46.2 | - | 38.5 | - | - | 43.7 | - | 43.5 |
| MoleculeSTM | 81.8 | - | 81.9 | - | - | 75.8 | - | 74.5 |
| MolCA w/o MTM | 77.3 | 97.7 | 77.3 | 97.5 | 38.1 | 81.5 | 37.3 | 81.6 |
| MolCA | **83.7** | **98.9** | **84.3** | **98.7** | **48.4** | **88.6** | **48.3** | **87.3** |

(c) Molecule-text retrieval performances in the MoMu dataset.

Table 7: Complete molecule-text retrieval performances on the datasets of PubChem324k, PCDes and MoMu. * denotes performance evaluated on the baseline's released checkpoint. † denotes result borrowed from (Su et al., 2022). Other models are trained on PubChem324k's pretrain subset.

experiment uses the same hyperparameters as the molecule captioning experiment. We append a text prompt "The molecule's IUPAC name is" after the molecule representations as the task description (*cf.*

| Model | Pretrain stage 1 | Pretrain stage 2 | BLEU-2 | BLEU-4 | ROUGE-1 | ROUGE-2 | ROUGE-L | METEOR |
|---|---|---|---|---|---|---|---|---|
| MolCA, Galac$_{1.3B}$ | ✗ | ✗ | 35.8 | 27.6 | 47.4 | 33.0 | 42.1 | 42.2 |
| MolCA, Galac$_{1.3B}$ | ✓ | ✗ | 37.7 | 29.5 | 49.2 | 34.9 | 43.8 | 44.5 |
| MolCA, Galac$_{1.3B}$ | ✓ | ✓ | **39.8** | **31.7** | **51.7** | **37.3** | **46.2** | **46.8** |

Table 8: Ablating MolCA's two pretrain stages by the task of molecule captioning in the PubChem324k dataset.

| Cross-Modal Projector | Representation Type | BLEU-2 | BLEU-4 | ROUGE-1 | ROUGE-2 | ROUGE-L | METEOR |
|---|---|---|---|---|---|---|---|
| - | 1D SMILES | 33.7 | 26.0 | 45.4 | 31.6 | 40.7 | 40.3 |
| Linear | 1D SMILES + 2D Graph | 34.6 | 27.8 | 46.7 | 32.9 | 41.9 | 41.2 |
| Q-Former | 1D SMILES + 2D Graph | **39.8** | **31.7** | **51.7** | **37.3** | **46.2** | **46.8** |

Table 9: Comparing different cross-modal projectors for molecule captioning on the PubChem324k dataset. All the compared methods apply LoRA fine-tuning on Galactica$_{1.3B}$.

(a) Samples of molecule captioning.

(b) Samples of IUPAC name prediction.

Figure 7: Examples of MolCA's molecule-to-text generation results. We highlight text snippets in blue that correctly describe the molecule structures in the predicted texts. To save space, some parts of texts are replaced by (...).

Figure 5).

**Molecule-Text Retrieval.** We use MolCA's checkpoint from pretrain stage 1 for retrieval without fine-tuning on any other datasets. This is similar to the setting of zero-shot retrieval in (Su et al., 2022; Liu et al., 2022b).

**Molecule Property Prediction.** Following (Hu et al., 2020), we fine-tune the models for 100 epochs and report the test performance selected by the valid set. For molecule classification, we attach a linear classifier after the mean pooling of the LM's hidden states of the last layer. We use the AdamW optimizer with a constant learning rate of 1e-4 and weight decay of 0.05. This experiment uses the same LoRA configuration as the molecule captioning experiment in the Pub-Chem324k dataset.

**Counting Functional Groups (FGs).** We use the molecules in PubChem324k's train set for fine-tuning and use the molecules in the valid set for evaluation. Following (Rong et al., 2020), we use RDkit (Landrum, 2013) to obtain the ground truth counts of FGs in every molecule. For each FG type, we employ a separate linear classifier to regress its numbers. Our model is trained using the Mean Square Error (MSE) loss function. Other settings,

including optimizer and LoRA, are the same as the Molecule Property Prediction experiment.

**Galactica.** Following the instructions in (Taylor et al., 2022), we wrap SMILES sequences with special tokens of [START_I_SMILES] and [END_I_SMILES] before feeding them into Galactica.

**PubChem324k Dataset.** Our dataset collection process follows the procedures described in (Liu et al., 2022b). The resulting dataset is larger due to the frequent updates made to the PubChem database (Kim et al., 2021). For each molecule in this website, we use the "description" field in its webpage as the corresponding text description. To avoid information leakage, we replace any common name or IUPAC name of the molecule at the beginning of texts with a text template (*i.e.,* "The molecule"). Detailed statistics of PubChem324k are presented in Table 6.

## C More Experimental Results

**Molecule-Text Retrieval**. Here we present MolCA's complete molecule-text retrieval performance on the PubChem324k, PCDes, and MoMu datasets. Following (Su et al., 2022), we report the performance of retrieval in a batch of 64 random

| Sample 1 | SMILES: C([C@@H]1[C@H]([C@@H]([C@H](C(O1)O)NS(=O)(=O)O)O)O)O[C@H]2[C@@H]([C@H](C(=C(O2)C(=O)O)O)O)O)OS(=O)(=O)O |
|---|---|
| **Ground truth** | The molecule is an amino disaccharide consisting of alpha-(...) joined in sequence by a (1->4) glycosidic bond. It is a disaccharide derivative, an oligosaccharide sulfate, a member of sulfamic acids, a monocarboxylic acid (...) |
| **1D SMILES** | The molecule is a disaccharide sulfate consisting of 2-acetamido-(...) joined in sequence by a (1->4) glycosidic bond. It is functionally related to a N-acetyl-D-glucosamine and a N-acetyl-D-galactosamine. |
| **1D SMILES + 2D Graph** | The molecule is a disaccharide that consists of 2-O-(...) residues joined in sequence by a (1->4) glycosidic bond. It is a disaccharide, an amino disaccharide, and a member of sulfamic acids. |
| **Sample 2** | SMILES: CCCCCCCCCCCCCCCCCCCCC(C(=O)O)O |
| **Ground truth** | The molecule is a long-chain fatty acid that is behenic acid substituted at position 2 by a hydroxy group. It is a 2-hydroxy fatty acid. It is functionally related to a docosanoic acid. It is a conjugate acid of a 2-hydroxybehenate. |
| **1D SMILES** | The molecule is a 2-hydroxy fatty acid that is the 2-hydroxy derivative of tetracosanoic acid. It is functionally related to a tetracosanoic acid. It is a conjugate acid of a 2-hydroxytetracosanoate. |
| **1D SMILES + 2D Graph** | The molecule is a 2-hydroxy fatty acid that is hexacosanoic acid substituted at position 2 by a hydroxy group. It is a long-chain fatty acid. It is functionally related to an hexacosanoic acid. It is a conjugate acid of a 2-hydroxyhexacosanoate. |

Table 10: Molecule captioning samples of MolCA (*i.e.,* 1D SMILES + 2D Graph) and its variant of using only 1D SMILES. We highlight text snippets in blue that correctly describe the molecule structures in the predicted texts. To save space, some parts of texts are replaced by (...).

samples and the performance of retrieval in the entire test set. As shown in Table 7, our conclusions align with those from Section 4.4: 1) MolCA consistently outperforms the baselines for molecule-text retrieval; 2) applying the MTM module for re-ranking is crucial for MolCA's molecule-text retrieval performances.

**Ablating the Pretrain Stages.** We conduct ablation studies on MolCA's two pretrain stages. As shown in Table 8, both the two pretrain stages have significant contributions to MolCA's molecule captioning performances.

**Ablating the Cross-Modal Projector.** We compare the performances of our selected cross-modal projector Q-Former and a linear cross-modal projector. For the linear cross-modal projector, we feed the node representations from the graph encoder to the base LM after the linear projector layer. We tune the weights of the graph encoder, linear projector, and the base LM's LoRA adapter. The experimental setting and hyperparameters are the same as those of MolCA. Table 9 shows the results. We can observe that: 1) Linear cross-modal projector underperforms Q-Former. We conjecture that a linear layer is suboptimal to bridge the modality gap between 2D molecules and 1D texts. This aligns with findings in the MME benchmark (Fu et al., 2023), where Q-Former-based methods (*e.g.,* BLIP-2, InstructBLIP (Dai et al., 2023), MiniGPT-

4 (Zhu et al., 2023)) outperform linear cross-modal projector based method (*e.g.,* LLaVA (Liu et al., 2023)). 2) Linear cross-modal projector slightly outperforms the SMILES-only baseline. We attribute this improvement to the usage of 2D molecular graphs, but the gains are limited because the linear projector is less effective.

**MolCA's Generation Results.** Figure 7 shows MolCA's molecule-to-text generation results. The two samples of molecule captioning is also presented in Table 10. Specifically, we compare MolCA (*i.e.,* 1D SMILES + 2D Graph) and its variant that is pretrained and fine-tuned using only 1D SMILES. We can observe that using both 1D SMILES and 2D graph leads to more accurate descriptions of molecule structures.

**Computational Cost.** We present the real-world training time of MolCA's three training stages in Table 11. All experiments are conducted on two NVIDIA A100 40 GB GPUs. Notably, we observe that the fine-tuning stage is affordable in terms of computational resources.

| Stage | Base LM | Dataset | Epochs | Time |
|---|---|---|---|---|
| Pretrain stage 1 | - | PubChem324k pretrain subset | 50 | 18.0h |
| Pretrain stage 2 | $Galac_{1.3B}$, freeze | PubChem324k pretrain subset | 10 | 9.0h |
| Pretrain stage 2 | $Galac_{125M}$, freeze | PubChem324k pretrain subset | 10 | 3.0h |
| Fine-tune stage | $Galac_{1.3B}$, LoRA ft | PubChem324k train subset | 100 | 6.0h |
| Fine-tune stage | $Galac_{125M}$, full ft | PubChem324k train subset | 100 | 1.5h |

Table 11: Compuational cost for MolCA's three stages.