# OpenReview forum: "MolCA: Molecular Graph-Language Modeling with Cross-Modal Projector and Uni-Modal Adapter"
_EMNLP/2023/Conference — EMNLP 2023 Main_

### Official Review · Reviewer_Cn2f · 2023-07-28

**Soundness:** 4

**Excitement:**

4: Strong: This paper deepens the understanding of some phenomenon or lowers the barriers to an existing research direction.

**Paper Topic And Main Contributions:**

Language Models have demonstrated impressive molecule understanding ability on various 1D text-related tasks. However, they inherently lack 2D graph perception.  In this work, the authors propose MolCA to bridge this gap. In details, they represent a graph encoder module to translate the representations of 2D graphs into 1D soft prompts in the text space, and a cross-modal projector to harness the power of existing large LMs for molecule-to-text generation. Besides, they integrate the LM with a uni-modal adapter LoRA to enable its efficient adaptation. The experimental results validate the effectiveness.

**Reasons To Accept:**

In this paper, authors propose MolCA to enables the LM to understand 2D graphs. This has some inspiration on how to better utilize other modal to improve the performance. Besides, the structure of this paper is coherent and compact, with the author's writing logic flowing smoothly. The graphs are clear, and the experimental results are abundant. Overall, it is well-received.

**Reasons To Reject:**

no

**Reproducibility:**

4: Could mostly reproduce the results, but there may be some variation because of sample variance or minor variations in their interpretation of the protocol or method.

**Reviewer Confidence:**

4: Quite sure. I tried to check the important points carefully. It's unlikely, though conceivable, that I missed something that should affect my ratings.

---

> ### Author Rebuttal · Authors · 2023-08-29
>
> Thank you for your thorough review and positive feedback on our submission. We are delighted to learn that you found our idea inspiring and our writing smooth and clear. To further enhance the quality of our submission, we will revise our draft based on each reviewer's comments and suggestions. Thanks again for your support!

---

### Official Review · Reviewer_JBDY · 2023-08-04

**Soundness:** 4

**Excitement:**

4: Strong: This paper deepens the understanding of some phenomenon or lowers the barriers to an existing research direction.

**Paper Topic And Main Contributions:**

This paper claims that although current large language model show impressive capacity in understanding molecule, the nature of LLMs to allow only 1D string as input restricted their abilities in further percept 2D topological structure of molecules. To address such problem, the paper propose a framework called MolCA. The proposed framework adopt Q-Former as cross-modal projector to the LLMs, and provide a pretrain pipeline to conduct cross-modal alignment. Empirical study shows excellent abilities of MolCA in improving LLMs' abilities of understanding molecules.

**Questions For The Authors:**

1. For training two Galactica models in your framework, what is the training cost (e.g., cost of time / gpus) for completing both pretrain stage 1 and stage 2?
2. Does MolCA has potential to extend to molecule classification and generation tasks? I notice in Table 5, although adding topological information, the improvement of performance is still limited, and is worse than some pretrained molecule encoder, which is trained only on 2D molecular graph data.
3. Can you provide results of MolT5 + MolCA and Galac with 1D smiles only?

**Reasons To Accept:**

1. The paper is clearly written and easy to follow.
2. The idea is novel and has great potential for expressing the power of LLMs into modeling molecules.
3. The authors conduct comprehensive experiments, including both on large language model with 1.3B parameters, and light-weight LM with only 125M parameters. This enhances the contribution of the proposed work.

**Reasons To Reject:**

1. It seems that the authors did not reveal the computational cost of training Galactica with MolCA's training pipeline, this hinder the understanding of the price in order to gain improvement claimed in the paper.

2. In their experiments, the authors only report results for MolT5 in 1D modeling and MolCA + Galac in 1D+2D modeling, making direct comparisons difficult.

**Reproducibility:**

4: Could mostly reproduce the results, but there may be some variation because of sample variance or minor variations in their interpretation of the protocol or method.

**Reviewer Confidence:**

3: Pretty sure, but there's a chance I missed something. Although I have a good feel for this area in general, I did not carefully check the paper's details, e.g., the math, experimental design, or novelty.

---

> ### Author Rebuttal · Authors · 2023-08-29
>
> We appreciate the time you've invested in reviewing our paper and are gratified to know that you consider our idea novel and has great potential. Your constructive insights are invaluable for the ongoing improvement of our research. Below, we address each of your comments in detail.
>
> > **Q1:** It seems that the authors did not reveal the computational cost of training Galactica with MolCA's training pipeline, this hinder the understanding of the price in order to gain improvement claimed in the paper. ... For training two Galactica models in your framework, what is the training cost (e.g., cost of time / gpus) for completing both pretrain stage 1 and stage 2?
>
> **Response:** Thank you for the advice on detailing the computational cost. We have revised our submission to report the computational costs for both pretraining and fine-tuning stages. The results are shown in the Table 1 below. Specifically, all experiments are conducted on two NVIDIA A100 40 GB GPUs.
>
> Notably, we observe the fine-tuning stage to be affordable in terms of computational resources. To further aid in the adoption of our method, we plan to release our pretrained models for public use.
>
> **Table 1:** Compuational cost for MolCA's three stages.
>
> | Stage            | Base LM                        | Dataset                     | Epochs | Time  |
> | ---------------- | ------------------------------ | --------------------------- | ------ | ----- |
> | Pretrain stage 1 | -                              | PubChem324k pretrain subset | 50     | 18.0h |
> | Pretrain stage 2 | Galac$_{\text{1.3B}}$, freeze  | PubChem324k pretrain subset | 10     | 9.0h  |
> | Pretrain stage 2 | Galac$_{\text{125M}}$, freeze  | PubChem324k pretrain subset | 10     | 3.0h  |
> | Fine-tune stage  | Galac$_{\text{1.3B}}$, LoRA ft | PubChem324k train subset    | 100    | 6.0h  |
> | Fine-tune stage  | Galac$_{\text{125M}}$, full ft | PubChem324k train subset    | 100    | 1.5h  |
>
> > **Q2:** In their experiments, the authors only report results for MolT5 in 1D modeling and MolCA + Galac in 1D+2D modeling, making direct comparisons difficult. ... Can you provide results of MolT5 + MolCA and Galac with 1D smiles only?
>
> **Response:** Thanks for the advice. We agree that these comparisons are crucial for a comprehensive understanding of MolCA's performance. For this purpose, we report performances of MolCA's variant that uses 1D SMILES only (*i.e.*, LoRA fine-tuned Galactica1.3B), and MolCA with different base LMs (*i.e.*, MolCA's MolT5 version).
>
> **Comparison with LoRA fine-tuned Galactica1.3B.** We have updated Table 5a in our submission to include Galactica1.3B's performance on the CheBI-20 dataset alongside the existing results on the PubChem324k dataset. Table 5a is shown below. We observe that adding 2D graph representations can significantly improve the performances for molecule captioning and IUPAC name prediction, which is consistent with the existing observations.
>
> **Table 5a:** Ablating the representation type on the PubChem324k dataset. All compared models fine-tune the base LM of Galactica1.3B.
>
> | Representation type                     | BLEU-2   | BLEU-4   | ROUGE-1  | ROUGE-2  | ROUGE-L  | METEOR   |
> | --------------------------------------- | -------- | -------- | -------- | -------- | -------- | -------- |
> | **Molecule  Caption, PubChem324k**      |          |          |          |          |          |          |
> | SMILES                                  | 33.7     | 26.0     | 45.4     | 31.6     | 40.7     | 40.3     |
> | Graph                                   | 35.7     | 27.4     | 47.3     | 32.3     | 41.8     | 42.0     |
> | SMILES+Graph                            | **39.8** | **31.7** | **51.7** | **37.3** | **46.2** | **46.8** |
> | **Molecule  Caption, CheBI-20**         |          |          |          |          |          |          |
> | SMILES                                  | 58.3     | 49.4     | 65.9     | 51.3     | 59.7     | 62.4     |
> | SMILES+Graph                            | **63.9** | **55.5** | **69.7** | **55.8** | **63.6** | **66.9** |
> | **IUPAC  name Prediction, PubChem324k** |          |          |          |          |          |          |
> | SMILES                                  | 70.7     | 60.7     | 68.6     | 46.2     | 61.7     | 71.5     |
> | SMILES+Graph                            | **74.6** | **66.1** | **70.5** | **49.1** | **64.2** | **73.0** |
>
> **Comparison with MolCA's MolT5 version.** We have updated Table 2a in our submission to include the performance of MolCA using MolT5-large as the base LM. We can observe that MolCA with MolT5-large consistently outperforms MolT5-large alone, showing MolCA's effectiveness in enabling LM to perceive 2D graphs.
>
> **Table 2a:** Performances (%) of molecule captioning on the PubChem324k datasets.
>
> | Model                     | BLEU-2   | BLEU-4   | ROUGE-1  | ROUGE-2  | ROUGE-L  | METEOR   |
> | ------------------------- | -------- | -------- | -------- | -------- | -------- | -------- |
> | **1D SMILES**             |          |          |          |          |          |          |
> | MolT5-small               | 14.8     | 8.5      | 26.5     | 13.5     | 23.6     | 18.5     |
> | MolT5-base                | 30.1     | 20.9     | 40.3     | 25.1     | 33.8     | 35.6     |
> | MolT5-large               | 30.2     | 22.2     | 41.5     | 25.9     | 34.8     | 36.6     |
> | **1D  SMILES + 2D Graph** |          |          |          |          |          |          |
> | MoMu-small                | 19.1     | 12.0     | 29.7     | 16.3     | 26.7     | 21.8     |
> | MoMu-base                 | 30.2     | 21.5     | 40.5     | 25.1     | 34.4     | 34.2     |
> | MoMu-large                | 31.1     | 22.8     | 41.8     | 25.7     | 36.7     | 36.2     |
> | MolCA, MolT5-large        | 33.7     | 27.0     | 49.7     | 35.6     | 44.4     | 42.4     |
> | MolCA, Galac125M          | 32.4     | 24.9     | 44.9     | 30.1     | 39.5     | 39.2     |
> | MolCA, Galac1.3B          | **39.8** | **31.7** | **51.7** | **37.3** | **46.2** | **46.8** |
>
> Your suggestions have been invaluable in refining our paper, and we believe these updates will provide a more comprehensive evaluation of MolCA's capabilities and performances.
>
> > **Q3:** Does MolCA has potential to extend to molecule classification and generation tasks? I notice in Table 5, although adding topological information, the improvement of performance is still limited, and is worse than some pretrained molecule encoder, which is trained only on 2D molecular graph data.
>
> **Response:** Thank you for your insightful question and the observation of MolCA's potential in the molecule classification task. In brief, MolCA does have the potential to be extended to molecule classification, and MolCA's base LM, Galactica, can be adapted for molecule generation, following [1].
>
> Now we elaborate in detail the possible methods to fully unleash the potential of MolCA and Galactica for molecule classification and molecule generation.
>
> * **Molecule Classification.** Compared to pretrained 2D molecular graph encoders, MolCA's advantage is the usage of an LM. Therefore, LM's abilities like in-context learning and chain-of-thought reasoning [2,3,4] can potentially improve molecule classification performance, especially in the few-shot setting and out-of-distribution setting. To fully unleash these abilities, we speculate that it is necessary to first fine-tune MolCA with molecule-centric instruction tuning data, which has been discussed in our response to Q2 of Reviewer YfAf.
> * **Molecule Generation.** MolCA's base LM, Galactica, can be fine-tuned for this task, following [1]. To achieve the optimal performance, we suggest first continuing pretrain Galactica on a large collection of molecule sequences, or weakly supervised (molecule, text) pairs, before applying it for text-based molecule generation.
>
> Regarding the performance of molecule classification, the current MolCA model is limited by the adopted molecular graph encoder and the molecule features. The most recent molecular graph encoders rely on molecules' 3D coordinates and carefully designed fingerprint features [5,6]. However, considering that molecule classification is not the main focus/contribution of this work, we leave the adoption of these advanced molecular graph encoders and molecule features to future exploration.
>
> In summary, we view these extensions as promising future directions. MolCA's ability to interpret the knowledge in text corpus provides it an advantage over uni-modal molecular methods.
>
>
>
> **Reference:**
>
> [1] Translation between Molecules and Natural Language. In EMNLP 2022.
>
> [2] The Flan Collection: Designing Data and Methods for Eﬀective Instruction Tuning. In Arxiv 2023.
>
> [3] Scaling Instruction-Finetuned Language Models. In Arxiv 2022.
>
> [4] InstructBLIP: Towards General-purpose Vision-Language Models with Instruction Tuning. In Arxiv 2023.
>
> [5] Uni-Mol: A Universal 3D Molecular Representation Learning Framework. In ICLR 2023.
>
> [6] Geometry-enhanced molecular representation learning for property prediction. In Nature Machine Intelligence 2022.

---

### Official Review · Reviewer_YfAf · 2023-08-05

**Soundness:** 5

**Excitement:**

4: Strong: This paper deepens the understanding of some phenomenon or lowers the barriers to an existing research direction.

**Missing References:**

There are some work focusing on Graph-Aware Language Model Pre-Training, e.g., Xie, Han, Da Zheng, Jun Ma, Houyu Zhang, Vassilis N. Ioannidis, Xiang Song, Qing Ping et al. "Graph-Aware Language Model Pre-Training on a Large Graph Corpus Can Help Multiple Graph Applications." arXiv preprint arXiv:2306.02592 (2023).

**Paper Topic And Main Contributions:**

The paper studies the open-ended molecule-to-text generation problem. The key point is to let language models understand the 2-dimensional molecule graph and translating the representations of 2D graphs into 1D soft prompts in the text space, and thus, bridging the gap between the graph encoder’s representation space and the LM’s input space. To this end, the author proposed MolCA and applies a three-stage training pipeline. In the first stage, a cross-modal projector and the graph-encoder are trained to extract the relevant molecule features to given the text, endowing the model with powerful molecule-text retrieval ability. In the second stage, the cross-modal projector is connected to a frozen LM and learn to produce soft prompts that the LM can understand. In the third stage, MolCA is fine-tuned for downstream generation tasks. Empirical results demonstrate that MolCA achieves SoTA results on various tasks, including molecule captioning and molecule-text retrieval.

**Questions For The Authors:**

How could we improve the Q-former to make MolCA sufficient for practical application? In addition to collect a larger pre-training corpus, do we need other tasks besides molecular captioning to align the representations of 2D molecular structure to text representation space?

**Reasons To Accept:**

Existing work exclusively utilizes 1D SMILES to represent molecules, while this paper proposed a a novel molecular language modeling method, which enable LMs to perceive molecules’ 2D graph representations. The model features a cross-modal Q-former to map representations of 2D graphs into the text space of LMs and applies an adapter for efficient downstream-finetuning.

The author evaluate the proposed method for molecule captioning and IUPAC name prediction  task on PubChem324k and CheBI-20, demonstrating a consistent superiority to the baseline methods.  Additionally, the proposed method also outperform the existing baseline on the Molecule-Text Retrieval task on three benchmark datasets. The author also conducts ablation studies to demonstrate that combing 2D graphs and 1D SMILES leads to improved performance.

**Reasons To Reject:**

Although the author ablate MolCA’s capability of counting FGs types inside molecules to demonstrate the LMs could understand 2D molecular structure, there is a lack of further in-depth analysis of how well the Q-former can attend to the corresponding graph structure conditioning on the textual input. For example, when the model caption a specific function of the given molecule, does the corresponding graph nodes have a high attention value?

**Reproducibility:**

4: Could mostly reproduce the results, but there may be some variation because of sample variance or minor variations in their interpretation of the protocol or method.

**Reviewer Confidence:**

3: Pretty sure, but there's a chance I missed something. Although I have a good feel for this area in general, I did not carefully check the paper's details, e.g., the math, experimental design, or novelty.

---

> ### Author Rebuttal · Authors · 2023-08-29
>
> Thank you for taking the time to review our paper. We're genuinely pleased to hear that you found the approach novel and the performance superior. The constructive feedback provided will undoubtedly help us further refine and improve our work. Here we provide a detailed response to each of your suggestions.
>
> > **Q1:** Although the author ablate MolCA’s capability of counting FGs types inside molecules to demonstrate the LMs could understand 2D molecular structure, there is a lack of further in-depth analysis of how well the Q-former can attend to the corresponding graph structure conditioning on the textual input. For example, when the model caption a specific function of the given molecule, does the corresponding graph nodes have a high attention value?
>
> **Response:** Thank you for your insightful comment. We appreciate your suggestion, but would like to clarify the inherent technical challenges in assessing the node-text attention map. To elucidate, let's consider two distinct stages:
>
> * **Pretrain Stage 1.** Q-Former establishes attentional relationships between graph nodes and query tokens (i.e., node-query attention map), as well as between query tokens and texts (i.e., text-query attention map). However, there isn't a direct attention connection between graph nodes and texts, thereby precluding direct access to the node-text attention map.
>
> *  **Pretrain Stage 2 and Fine-tune Stage.** During these stages, Q-Former processes graph node inputs, without the text inputs. As a result, the graph nodes lack the capacity to attend to texts within Q-Former's framework. Although the base LM has text inputs, its 2D graph inputs are not provided on a node-wise basis, making it difficult to evaluate attention values between texts and nodes.
>
> We fully recognize the importance of node-text attention map and will explore it in future work, to resolve the outlined challenges.
>
>
>
> > **Q2:** How could we improve the Q-former to make MolCA sufficient for practical application? In addition to collect a larger pre-training corpus, do we need other tasks besides molecular captioning to align the representations of 2D molecular structure to text representation space?
>
> **Response:** Thanks for the insightful question on improving practical application performance. As you rightly pointed out, collecting a larger pre-training corpus is a straightforward method for performance improvement. And we agree that a larger dataset alone may not be sufficient. Beyond that, we foresee two other potential methods for improvement:
>
> * **Molecular Graph-Language Modeling with Multiple Molecules.** Simultaneously modeling multiple molecules requires the LM to understand and process the interactions and relations between multiple molecule inputs, an ability that can hardly be learned by modeling a single molecule. Properly cultivating this ability can potentially benefit downstream tasks that require reasoning across multiple molecules, such as chemical reaction prediction [1] and retrosynthesis [2].
>
> * **Molecule-centric Instruction Tuning.** It is possible to improve MolCA's performance by aligning its generation results to human preference using instruction tuning [3,4,5]. We speculate that molecule-centric instruction tuning can be achieved by fine-tuning MolCA on a diverse set of molecule-related text generation tasks. It can potentially improve MolCA in two distinct ways:
>   * It augments MolCA's ability to interact with humans. For example, if molecule-centric question-answering data can be included in instruction tuning, we can let MolCA's generation results focus on specific molecule properties by raising the corresponding question, rather than merely presenting all relevant information.
>   * If chain-of-thought annotation data can be included, it can potentially improve MolCA's ability of few-shot learning [3,4]. For pure texts, chain-of-thought annotations decompose each task into a sequence of sub-tasks [6]. Through learning to perform these sub-tasks, LMs have potentials to generalize to new tasks using a few examples [6].
>
>
>
> > **Q3:** There are some works focusing on Graph-Aware Language Model Pre-Training, e.g., Xie, Han, Da Zheng, Jun Ma, Houyu Zhang, Vassilis N. Ioannidis, Xiang Song, Qing Ping et al. "Graph-Aware Language Model Pre-Training on a Large Graph Corpus Can Help Multiple Graph Applications." arXiv preprint arXiv:2306.02592 (2023).
>
> **Response:** Thank you for kindly pointing out the concurrent reference by (Xie et al., 2023). Indeed, the work is relevant as it also focuses on joint learning of 2D graphs and 1D texts. We did not include this concurrent work in our initial submission due to its public availability on June 5, 2023, which fell within the same month of EMNLP's submission deadline. We have promptly addressed this oversight by incorporating this work into our revised related work section.
>
> In comparison with the referenced work, we identify two key distinctions that set our work apart:
>
> * **Different Domains.** We focus on molecular graphs while the reference focuses on social networks. The graphs from different domains differ in both structure and scale, which calls for different technical solutions.
>
> * **Different Purposes.** We aim to enable an LM to understand 2D molecular graphs. In contrast, the referenced work leverages the LM as a text feature extractor for Graph Neural Networks.
>
> We believe these clarifications highlight the uniqueness of our approach, and we appreciate the opportunity to enhance our paper with the inclusion of this relevant reference.
>
>
>
> **Reference:**
>
> [1] Chemical-reaction-aware molecule representation learning. In ICLR 2022
>
> [2] Retrosynthetic reaction pathway prediction through neural machine translation of atomic environments. In Nature Communications 2022.
>
> [3] The Flan Collection: Designing Data and Methods for Eﬀective Instruction Tuning. In Arxiv 2023.
>
> [4] Scaling Instruction-Finetuned Language Models. In Arxiv 2022.
>
> [5] InstructBLIP: Towards General-purpose Vision-Language Models with Instruction Tuning. In Arxiv 2023.
>
> [6] Chain-of-Thought Prompting Elicits Reasoning in Large Language Models. In NeurIPS 2022.

---

### Meta-Review · Area_Chair_pdkD · 2023-09-19

**Recommendation:** 5

**Metareview:**

The paper addresses the problem of making LLMs understand 2d molecular structure They propose MoICA which contains (i) a cross-modal projector to connect a graph encoder's representation space and an LM's text space and (ii) a uni-modal adapter to help the LLM to adapt to downstream tasks. They show that MolCA significantly outperforms existing methods on tasks of molecule captioning, IUPAC name prediction, and molecule-text retrieval.

I found this to be a very interesting paper (mainly because it deals with very unique modalities). The reviewers are very positive about this work and have not raised any serious concerns. They agree that the approach is novel and the evaluation is comprehensive. There was a concern about the comparison with related word was not directly comparable but this has been addressed in the response.

Overall, the paper is sound and exciting. I request the authors to include the additional experiments and clarifications provided during the rebuttal into the final version of the paper.

---

### Decision · Program_Chairs · 2023-10-07

**Decision:**

Accept-Main

**Comment:**

The paper addresses the problem of making LLMs understand 2d molecular structure They propose MoICA which contains (i) a cross-modal projector to connect a graph encoder's representation space and an LM's text space and (ii) a uni-modal adapter to help the LLM to adapt to downstream tasks. They show that MolCA significantly outperforms existing methods on tasks of molecule captioning, IUPAC name prediction, and molecule-text retrieval.

I found this to be a very interesting paper (mainly because it deals with very unique modalities). The reviewers are very positive about this work and have not raised any serious concerns. They agree that the approach is novel and the evaluation is comprehensive. There was a concern about the comparison with related word was not directly comparable but this has been addressed in the response.

Overall, the paper is sound and exciting. I request the authors to include the additional experiments and clarifications provided during the rebuttal into the final version of the paper.